# Direct Quantification of Drug Loading Content in Polymeric Nanoparticles by Infrared Spectroscopy

**DOI:** 10.3390/pharmaceutics12100912

**Published:** 2020-09-23

**Authors:** Guzmán Carissimi, Mercedes G. Montalbán, Gloria Víllora, Andreas Barth

**Affiliations:** 1Department of Chemical Engineering, Faculty of Chemistry, University of Murcia, 30100 Murcia, Spain; guzmanaugusto.carissimin@um.es (G.C.); mercedes.garcia@um.es (M.G.M.); gvillora@um.es (G.V.); 2Department of Biochemistry and Biophysics, Stockholm University, SE-106 91 Stockholm, Sweden

**Keywords:** silk fibroin, nanoparticles, drug loading content, quantification, infrared spectroscopy, FTIR spectroscopy, nanotechnology, nanomedicine, drug delivery, controlled release

## Abstract

Nanotechnology has enabled the development of novel therapeutic strategies such as targeted nanodrug delivery systems, control and stimulus-responsive release mechanisms, and the production of theranostic agents. As a prerequisite for the use of nanoparticles as drug delivery systems, the amount of loaded drug must be precisely quantified, a task for which two approaches are currently used. However, both approaches suffer from the inefficiencies of drug extraction and of the solid-liquid separation process, as well as from dilution errors. This work describes a new, reliable, and simple method for direct drug quantification in polymeric nanoparticles using attenuated total reflection Fourier transform infrared spectroscopy, which can be adapted for a wide variety of drug delivery systems. Silk fibroin nanoparticles and naringenin were used as model polymeric nanoparticle carrier and drug, respectively. The specificity, linearity, detection limit, precision, and accuracy of the spectroscopic approach were determined in order to validate the method. A good linear relation was observed within 0.00 to 7.89% of naringenin relative mass with an R^2^ of 0.973. The accuracy was determined by the spike and recovery method. The results showed an average 104% recovery. The limit of detection and limit of quantification of the drug loading content were determined to be 0.3 and 1.0%, respectively. The method’s robustness is demonstrated by the notable similarities between the calibrations carried out using two different equipment setups at two different institutions.

## 1. Introduction

The use of nanoparticles as drug delivery systems is already a reality, with more than 50 nanomedicines approved by the Food and Drug Administration (USA) in April 2016, and more to come in the foreseeable future [1]. The use of nanoparticles as delivery systems may provide numerous benefits [2,3,4], including enhanced drug delivery to tumor cells [5,6], controlled release and improved biodistribution [7], all of which result in a reduction of drug side effects [8,9]. Among polymeric nanoparticles used as delivery systems, silk fibroin-based nanoparticles (SFN) from the silkworm *Bombyx mori* have shown promising results, partly due to their biodegradability, biocompatibility and nontoxicity [10,11,12,13,14,15,16]. Their ability to penetrate biological membranes and their high drug loading capacity mean that SFN can greatly improve therapeutic treatments [17,18]. Furthermore, the SFN surface can be functionalized by attaching recognition molecules for targeted delivery, such as peptides or other small molecules [19,20].

However, if nanoparticles are to be used as drug delivery systems, the amount of loaded drug must be precisely quantified. To date, two approaches are mostly used for this task. They are based on a direct and indirect measurement of the drug loading content (*DLC*) in the nanoparticle. In the indirect approach [21,22,23,24,25], a fresh loading solution of the drug is prepared and measured before adding the nanoparticles. The suspension is stirred, and at the end of the loading process [13], the particles are separated from the loading solution, typically by centrifugation or filtration, and the remaining drug in the supernatant solution is measured. Finally, the amount of drug in the nanoparticles is calculated by a mass balance. However, as high drug concentrations are normally used for the loading solution, large dilution factors are needed to keep the analyte signal in the equipment’s operating range, when conventional UV-visible spectrometers or high-pressure liquid chromatographs are used. This adds uncertainty, and thus, lowers the accuracy and the precision of the determined drug content. Furthermore, this approach cannot be used for independent control of the drug content of the final product, because only the producer will have access to the loading solution. Further still, the drug content may change upon storage, in which case an analysis of the loading solution will be meaningless.

As for the direct approach [26,27,28,29], the drug from the loaded particles is first extracted with a suitable solvent; then, particles are sedimented and the drug in the supernatant is measured. However, it has been shown that separation methods such as filtration or centrifugation are not always effective for separating nanoparticles from the liquid phase, leading to errors in spectroscopic measurements [30,31]. Even so, extracting the total loaded drug is difficult and hard to verify. Moreover, both methods assume that the drug absorption is in the UV-visible range, and that there is no interference with the nanoparticles and their dissociated components that may remain in dispersion after the separation process.

For these reasons, this work develops a reliable and simple method for directly quantifying drugs in nanoparticle dispersions/suspensions which can be adapted for use with other types of nanostructures. To the best of our knowledge, this is the first time that attenuated total reflection Fourier transform infrared (ATR-FTIR) spectroscopy has been used for the direct quantification of the drug-load of nanoparticles. FTIR spectroscopy is a well-established technique within the pharmaceutical industry with applications ranging from qualitative to quantitative analyses [32,33,34,35,36,37]. In the latter, its major advantage over UV-visible spectroscopy is that practically all compounds absorb IR light and can thus be analyzed. In addition, the IR spectrum typically shows a large number of characteristic absorption bands with limited overlap because they are distributed over a wide spectral range.

As a model drug, we used SFN loaded with naringenin (NAR), a natural flavonoid [38,39], which has recently demonstrated its good compatibility with silk fibroin [16]. NAR has received increasing attention in medicine due to its free-radical scavenging [40], anti-inflammatory [41] and anticancer properties [42,43,44,45,46]. However, NAR quantification remains a problem, especially when loaded into protein-based nanoparticles, as its absorption maximum at 289 nm overlaps the absorption of aromatic amino acids of proteins. The robustness of the proposed method is demonstrated by consistent results from two different laboratories in a collaboration between the University of Murcia (Spain) and Stockholm University (Sweden). The two institutions used different equipment, and their respective latitudes are significantly different, meaning that the performance of the method can be compared at different temperatures and in different climate conditions.

## 2. Materials and Methods 

### 2.1. Materials

The silk fibroin (SF) used in this work was extracted from *Bombyx mori* silk cocoons kindly provided by Instituto Murciano de investigación y Desarrollo Agrario y Alimentario (Murcia, Spain). The SF was purified as reported elsewhere [47]. In brief, cocoons were shredded in a mill and filter through a 1 mm stainless still mesh and later boiled for 2 h in a 0.2 N Na_2_CO_3_ solution to remove sericin, waxes and other impurities. Following this, the SF was collected in a 0.1 mm pore size strainer and washed thoroughly with warm water to remove any unbounded sericin. Finally, the SF was air-dried in a fume hood until constant weight was achieved. NAR was purchased from Sigma-Aldrich (Madrid, Spain) and was used as received. Purified water from a Millipore Direct-Q1 ultrapure water system (18.2 MΩ·cm at 25 °C; Billerica, MA, USA) was used throughout the experimentation.

### 2.2. Preparation of Stock Nanoparticle Dispersion and Drug Solution 

SFN were prepared as described elsewhere [13,16]. The freeze-dried SFN were dispersed in Milli-Q water by ultrasonication probe to a final concentration of 20 mg/mL. A stock NAR solution was prepared in ethanol by stirring vigorously until complete dissolution at a final concentration of 15 mg/mL. Throughout this work, all the solutions containing NAR were always kept protected from light.

### 2.3. Calibration Samples for NAR in SFN 

To calibrate the ATR-FTIR method, six samples were prepared with NAR relative masses (*NAR%*) of 0.00, 1.06, 2.10, 4.11, 6.04 and 7.89%, calculated as described in Equation (1), where *m*_NAR_ and *m*_SFN_ are the masses of NAR and SFN in the NAR/SFN dispersion, respectively.
(1)$$NAR\%=\frac{m_{NAR}}{m_{NAR}+m_{SFN}}\times 100$$

Samples were prepared by adding a fixed volume of SFN stock dispersion, followed by a variable volume of NAR stock solution and, finally, a variable volume of pure ethanol so that the total volume was 100 µL. The pure ethanol was added to maintain a constant proportion of 30% ethanol, thus ensuring that SFN remained totally dispersed and the NAR totally dissolved. The SFN mass was kept constant in all the samples, and only the NAR mass was increased between samples. These calibration samples were used to measure the NAR infrared absorption spectrum in the presence of SFN, which contained contributions from free and loaded NAR. The samples were prepared freshly every day and measured in triplicate at Stockholm University and University of Murcia, giving a total of six calibration lines, all of which were obtained on different days.

### 2.4. NAR-Loaded SFN Samples

To test the drug detection and quantification capabilities of ATR-FTIR spectroscopy, three samples of SFN were loaded with NAR by SFN incubation in drug solutions with NAR concentrations of 0.30, 1.2 and 4.5 mg/mL for 24 h. Samples were named SFN_NAR_-0.3 mg/mL, SFN_NAR_-1.2 mg/mL and SFN_NAR_-4.5 mg/mL, according to the concentration of the drug in the loading solution, and were prepared in triplicate on different days. The final SFN concentration in each sample was 14 mg/mL, and the final volume of the loading sample was 100 µL. As for the calibration samples, the loading drug solutions contained 30% ethanol, thus ensuring that SFN remained totally dispersed and the NAR totally dissolved. Loading SFN/NAR dispersions were sonicated before starting the incubation period to ensure homogeneity. After the incubation time, the nanoparticles were centrifuged for 20 min at 12.100× *g*, after which the supernatant with the unloaded/unbounded drug was removed. Subsequently, the nanoparticles were washed with 30% EtOH to remove the unbound drug. Finally, drug-loaded nanoparticles were re-dispersed in milli-Q water by ultrasonication before quantification of drug loading.

The drug loading content (*DLC*) and entrapment efficiency (*EE*) of the SFN_NAR_ were calculated according to the following expressions:(2)$$DLC\left(\%\right)=\frac{weight\;of\;NAR\;in\;the\;nanoparticle}{weight\;of\;SFN_{NAR}}\times 100$$
(3)$$EE\left(\%\right)=\frac{weight\;of\;NAR\;in\;the\;nanoparticle}{weight\;of\;NAR\;in\;the\;loading\;solution}\times 100$$

In Equation (2), the weight of *SFN_NAR_* is the sum of the masses of NAR and SFN in the loaded nanoparticles, making it analogous to Equation (1). Thus, *DLC* can be directly read out from our calibration line.

### 2.5. Spectra Acquisition and Data Pretreatment

The infrared absorption spectra measured at Stockholm University were recorded on a Vertex 70 Fourier-transform infrared spectrometer (Bruker Optics, Ettlingen, Germany) equipped with a liquid nitrogen-cooled Mercury-Cadmium-Telluride detector and a 1-reflection, 45° angle of incidence diamond ATR accessory (Platinum, Bruker Optics, Ettlingen, Germany). The equipment was continuously purged with CO_2_-reduced, dry air. Each measured spectrum was averaged from 300 scans at a data collection rate of 9.38 scans per second.

The infrared absorption spectra measured at the University of Murcia were recorded on an iS5-Nicolet Fourier-transform infrared spectrometer (Thermo Fischer Scientific, Waltham, MA, USA) equipped with a Deuterated Triglycine Sulfate detector and a 1-reflection, 45° angle of incidence diamond ATR accessory (iD7 ATR module, Thermo Fischer Scientific, Waltham, MA, USA). Each measured spectrum was the average of 32 scans at a data collection rate of 0.47 scans per second.

In both cases, interferograms were recorded at a resolution of 2 cm^−1^ with a zero-filling factor of 2, and Fourier-transformed using the Blackman-Harris 3-term apodization function. A background spectrum without a sample with the same number of scans was collected before each measurement. The spectra of the samples were acquired in the form of dry films by placing 2.5 µL of the nanoparticle dispersion on top of the ATR-crystal and drying it under a gentle stream of nitrogen before measuring as described above. The film thickness may have dramatic effects on the resultant ATR spectrum [48,49]; to obtain reproducible results, the thickness of the dry film must be greater than the penetration depth of the evanescent wave into the sample. To ensure this, we placed a piece of plastic tape on top of the dry film sample, applied pressure with the ATR press, and then took the measurement. The absence of tape signals in the spectrum guaranteed that the penetration depth of the evanescent wave was smaller than the thickness of the dry film, and hence, that the amount of sample placed on the ATR crystal was sufficient. This procedure was performed only once; then, the volume and concentration were kept constant for the rest of the experimentation.

As a pretreatment for drug quantification, each spectrum was truncated to the region between 1350 and 900 cm^−1^; then, a baseline was subtracted by tracing straight lines between the minima of the SFN spectrum at 1350, 1316, 1185, 1134 and 900 cm^−1^. Finally, the absorbance scale was normalized to 1 for the amide III band of silk fibroin (1229 cm^−1^). The 1085 cm^−1^ band of NAR was chosen to quantify the NAR content in SFN because of its high absorption coefficient and its insensitivity to NAR incorporation into SFN (see Results and Discussion). The calibration samples were used to establish a correlation between the amount of NAR and the absorbance at 1085 cm^-1^ as described in Results and Discussion.

## 3. Results and Discussion

### 3.1. Specificity of ATR-FTIR Spectroscopy for Identifying Drugs in SFN 

The specificity of ATR-FTIR spectroscopy was tested by performing a spectral coincidence match where the spectrum of SFN_NAR_-0.3 was compared with that of pure NAR. Before comparing the two spectra, a SFN spectrum was subtracted from the SFN_NAR_-0.3 spectrum. As a consistent subtraction of both the amide I and amide II band of SFN was not possible, our compromise criterion for spectral subtraction was to obtain a positive and a negative band at the positions of the amide I and II bands, respectively, while trying to maximize signals from NAR. The OMNIC software V9.9.473 (Thermo Fisher Scientific, Waltham, MA, USA) was used to compare the resulting spectrum and the NAR spectrum, both of which can be seen in Figure 1. The region from 1700 to 1450 cm^−1^ was distorted by the subtraction of the SFN spectrum due to the presence of the intense amide I and II bands, and was excluded from the spectral matching. Only the region from 1400 to 1000 cm^−1^ highlighted by the gray area in the figure was used for the spectral match. The OMNIC software found a coincidence match of 86%. This result demonstrates that the drug in the nanoparticles could be identified, even at low concentrations.

The result suggests that this approach could also be used with other proteins as drug carriers, because the absorption of the proteins is relatively weak below 1400 cm^−1^, where the IR spectrum is highly characteristic of the structure of the molecules. Drugs with an appreciable absorption in this fingerprint region of the IR spectrum can, therefore, be identified.

### 3.2. Identification of the Best Marker Band for NAR in SFN_NAR_ Samples 

The first step in the quantification process was to identify a strong signal from NAR in the IR spectrum, with low interference from the silk fibroin matrix, to be used to generate a calibration curve. In Figure 2a, the spectra of SFN and NAR are presented from 1750 to 900 cm^−1^. The SFN spectrum presents the expected amide I, II and III bands at 1620, 1514 and 1229 cm^−1^, respectively, for silk fibroin protein [50,51,52]. 

The NAR spectrum presents strong absorption bands positioned at 1633, 1609, 1594, 1169, 1156, 1085 and 1065 cm^−1^. The first three bands arise from vibrations comprising different combinations of C=O stretching, ring C-C stretching and OH bending vibrations [53,54]. The reader can refer to Figure 3 for the structure of NAR. These three bands overlap with the strong amide I band of SFN, making NAR detection from these signals rather challenging; therefore, they were not taken into account here.

Bands at 1169 and 1156 cm^−1^ arise from a combination of OH and CH_2_ bending vibrations [54]. Neither these bands were considered for quantification, as hydroxyls may interact with the protein matrix from SFN, which will cause band shifts upon binding of NAR to SFN. Indeed, it can be seen in Figure 2b that the 1156 cm^−1^ absorption band of NAR is upshifted by 6 cm^−1^ in the spectrum of NAR-loaded SFN (SFN_NAR_-1.2). As stated above, this is probably due to hydrogen bonding with the protein matrix. This fact renders the band unsuitable for quantification purposes because of its ambiguous spectral position, and because of a possible impact of hydrogen bonding on the band intensity. Finally, the bands at 1085 and 1065 cm^−1^, which arise from aromatic skeletal vibrations of the flavone ring [54] of NAR, can also be detected in the SFN_NAR_ spectrum and are good candidates for quantification marker bands, as no noticeable shift was detected between free and loaded NAR. However, the latter band is close to a maximum of SFN absorption, which makes it difficult to distinguish between NAR and SFN absorption. In contrast, the strong and sharp 1085 cm^−1^ band is located on a slope of the SFN spectrum and can, therefore, be distinguished from it and used for quantification. For this reason, we decided to use the 1085 cm^-1^ absorption band for quantification.

The next step consisted of identifying a band from the protein matrix that can be used for normalization. At this point, it is also important to check for shifts in the absorption bands from the polymer matrix, which could be induced by interaction with the drug. For example, it can be seen from Figure 1 that the subtraction of SFN absorption from the SFN_NAR_-0.3 absorption spectrum did not cancel the absorbance in the amide I and II regions of protein absorption (note that the NAR absorption is much less at the concentrations used for drug-loaded SFN than indicated by the NAR spectrum in Figure 1; therefore, the NAR contribution is negligible in the amide I and II spectral ranges). This is due to a mismatch of the amide I and II bands of pure and drug-loaded SFN, which was possibly produced by interactions from the drug and polymer matrix. Here, the amide III band at 1229 cm^−1^, which is the strongest signal from SFN in the 1350-900 cm^−1^ region and which did not show any shift, was used for normalization.

With the quantification band from the drug and the normalization band from the nanoparticle matrix identified, the spectra needed to be baseline corrected before further operations. As baseline points, we selected points at the edges of the spectra (1350 and 900 cm^−1^), the minima on both sides of the normalization band (1316 and 1185 cm^−1^) and one extra point close to the quantification band at a minimum of the SFN spectrum (1134 cm^−1^). These are shown as orange dots in Figure 2.

### 3.3. Calibration Curve

The next step in the quantification process was to obtain the calibration curve between drug load and the absorbance at 1085 cm^−1^. This was achieved by analyzing the calibration samples with known amounts of SFN and NAR. An example of a series of baseline-corrected and normalized spectra of calibration samples containing an increasing amount of NAR is presented in Figure 4a, in which the increase in absorbance at 1085 cm^−1^ as the NAR concentration increases can be clearly seen. The fittings of the six independent calibration series, i.e., three obtained at the University of Murcia and three obtained at Stockholm University, are shown in Figure 4b. In Figure 4c, one calibration curve was fitted to all individual data points. This calibration curve was later used for quantification purposes in this work. The signal at 0 *NAR%* in Figure 4b,c stems from SFN absorption. The data were fitted by linear regression using the least-squares method. For the final calibration curve shown in Figure 4c, the Y-intercept was constrained to be equal to the average value of absorbance at 0 *NAR%* of the six samples (this value remained the same even when we included the ten additional measurements of SFN blanks). We used this approach to minimize the error for determining low NAR loads, which depends strongly on the starting point of the calibration curve. The fitting parameters are collected in Table 1. 

As a proof of concept, we performed the same calibration by plotting the negative intensity of the second derivative at 1085 cm^−1^ vs *NAR%*, as shown in Figure 5. As shown in Figure 5a, the peak at 1085 cm^−1^ is clearly resolved and a good linear relation was obtained (Figure 5b, R^2^ = 0.988). This procedure presents several advantages, as no baseline subtraction is needed and it could be used to resolve overlapping peaks in cases in which they are more difficult to analyze than those presented here. However, an analysis of second derivative spectra requires spectra with low noise, and here, the Mercury-Cadmium-Telluride detector in the instrument at Stockholm Unversity proved to be beneficial compared to the Deuterated Triglycine Sulfate detector in the instrument at the University of Murcia, although we chose a longer spectrum recording time for latter.

### 3.4. Parameters for Method Validation

#### 3.4.1. Range and Linearity of the Calibration Curve

The linearity of the calibration data was good, as indicated by the *R^2^* values for the individual calibration lines and for the single calibration with all data points (Table 1), which were equal to or above 0.973. Also, a visual inspection of the residual plot (data not shown) demonstrates a linear correlation between analytical response and drug concentration over the range of 0–7.89 *NAR%*.

#### 3.4.2. Robustness of the Method

The robustness of the analytical method is illustrated by the strong similarities between the six calibration lines in Figure 4b measured on different days, in different laboratories and with different spectrometers. For instance, the SU Bruker spectrometer used a Mercury-Cadmium-Telluride detector and was continuously purged with CO_2_-reduced, dry air, whereas the University of Murcia Thermo Fisher Scientific spectrometer used a Deuterated Triglycine Sulfate detector and was not purged. Despite this, an analysis of covariance performed as described elsewhere [55] showed no significant differences between the slopes or Y-intercepts of the lines generated at the two institutions with *P*-values of 0.92 and 0.18, respectively. The agreement demonstrates, among other things, that water vapor does not interfere with the measurements because the selected NAR band is outside the main region of water vapor absorption. The relative standard deviations calculated from the average and standard deviation of all slopes and Y-axis intercepts were 3.8 and 5.2%, respectively.

#### 3.4.3. Limit of Detection, Limit of Quantification and Reproducibility (Precision) of the Method

As defined by the IUPAC Gold Book [56,57], the limit of detection (*LOD*) and of quantification (*LOQ*), expressed as concentration or quantity, is the smallest measured value (*A*) that can be detected or quantified, respectively, with reasonable certainty for a given analytical procedure. *A_LOD_* and *A_LOQ_* are given by Equations (4) and (5), respectively [56,57]:(4)$$\begin{array}{c} \begin{array}{c} \begin{array}{c} \begin{array}{c} A_{LOD}=\bar{A}+3.3\sigma \end{array} \end{array} \end{array} \end{array}$$
(5)$$\begin{array}{c} \begin{array}{c} \begin{array}{c} \begin{array}{c} A_{LOQ}=\bar{A}+10\sigma \end{array} \end{array} \end{array} \end{array}$$
where $\bar{A}$ and σ are the average and standard deviation, respectively, of the absorbance measurements at 1085 cm^−1^ for ten independent blank SFN samples, five of which were made at the University of Murcia and the other five at Stockholm University. The number of ten samples and the numerical factors of 3.3 and 10 were suggested by IUPAC according to normality and a confidence level of *α* = 0.05 for *A_LOD_* and *A_LOQ_*, respectively. The reproducibility of the method is given by the relative standard deviation of the blank measurements calculated by Equation (6):(6)$$\begin{array}{c} \begin{array}{c} \begin{array}{c} \begin{array}{c} \begin{array}{c} Reproducibility\;\left(\%\right)=\frac{\sigma }{\bar{A}}\times 100 \end{array} \end{array} \end{array} \end{array} \end{array}$$

The ten spectra used and their absorbance at 1085 cm^−1^ are presented in Figure 6, and our evaluation thereof is summarized in Table 2. The results for *DLC_LOD_*, *DLC_LOQ_* and reproducibility are also collected in Table 2. The first two were calculated by inserting *A_LOD_* and *A_LOQ_* into the equation for the final calibration line. This result shows the high sensitivity and precision of the analytical method developed here.

#### 3.4.4. Accuracy of the Method

The accuracy of the proposed method was evaluated by the spike and recovery method [57,58]. First, a calibration sample with 4.11 *NAR%* relative mass was prepared and divided into two 200 µL aliquots. One aliquot was spiked with 5 µL of a 15 mg/mL NAR solution in EtOH to achieve a final 6.51 *NAR%* relative mass, while 5 µL of EtOH was added to the other sample. The infrared spectra of the spiked and unspiked samples were recorded immediately and the respective *DLC*s determined from the absorbance at 1085 cm^−1^ as described. The accuracy was expressed as the recovery % of the spiked sample, as stated in equation 7. The procedure was repeated four times and the results are shown in Table 3. A variation of 4% on average from the expected values indicated that the analytical procedure was accurate, and that there were no matrix effects. The small error was within the expected errors of pipetting, weighing and absorbance measurement.
(7)$$Recovery\;\left(\%\right)=\frac{DLC_{spiked}-DLC_{unspiked}}{NAR{\%}_{added}}\times 100$$

### 3.5. Drug Quantification in NAR-Loaded SFN

The *DLC* and *EE* (entrapment efficiency) of the NAR-loaded SFN samples measured with this method are presented in Table 4. An increase in the NAR concentration of the loading solution led to a nonlinear increase in *DLC*, with most of the loading activity taking place at a low NAR concentration. This has been reported before and was attributed to the saturation of the drug binding sites on the surface of the nanoparticles [15]. By contrast, the EE decreased linearly when the NAR concentration increased.

Note that the nature of the drug can be identified by infrared spectroscopy, even for the nanoparticles with the lowest drug load (Figure 1, Section 3.1, Specificity of ATR-FTIR spectroscopy for identifying drugs in SFN), although its determined *DLC* lies between the *DLC_LOQ_* and *DLC*_LOD_.

The relative standard deviations of the measured *DLC* values were strikingly low for measurements above the *DLC_LOQ_* of 1.0% (Table 4), especially when compared to UV-visible spectroscopy and high-performance liquid chromatography measurements used by the direct or indirect methods mentioned in the Introduction. Based on our experience, the coefficient of variations for these methods tend to be much larger, ranging from several percent to over 10%, mostly due to large dilution of the supernatants needed (up to 1:500) and unreliable phase separation through centrifugation (data not shown). These large and sometimes unacceptable deviations were the primary motivating factors for this work in the first place. The problems involved will be discussed below, using the SFN_NAR_-4.5 mg/mL sample as a example. To calculate its *DLC* by the indirect approach, particles needed to be separated from the initially 4.5 mg/mL NAR solution after the loading phase. Assuming complete removal and an EE of 15% (close to what was obtained here; see Table 4), the drug concentration in the supernatant should be close to 3.8 mg/mL, corresponding to 85% of the total drug feed. Such a high concentration cannot be directly measured with conventional UV-visible spectrometers. In the case of NAR dissolved in EtOH, an absorbance of 1.0 was achieved with a concentration of 0.015 mg/mL at 289 nm (10 mm path length cuvette). This means that in order to bring the concentration of a 3.8 mg/mL sample into a suitable concentration range for UV-Visible spectroscopy, a 1:250 dilution is required, which carries a high degree of error. Assuming a two-step dilution—corresponding to four pipetted volumes—and assuming a pipetting error of 2%, the final concentration error was close to 4%. The situation improved at lower concentrations of the loading solution, where the EE was higher. For example, the SFN_NAR_-0.3 sample with an EE close to 50% required only a 1:10 dilution. However, only low *DLC*s were achieved with such low drug concentrations, whereas high *DLC*s are the aim of drug producers. Thus, the dilution error of the indirect approach is highest under the conditions that are most interesting from a drug manufacturing point of view, whereas they are lowest for the direct method presented in this work. Furthermore, the separation of SFN from the NAR loading solution is ineffective. Control samples containing 1 mg/mL of SFN dispersed in 30% EtOH had an absorbance greater than 0.100 at 289 nm after attempted separation of SFN from the solution phase by centrifugation for 30 min at 13,400 rpm. The absorbance stems from the nanoparticles and it would add up to the absorbance of the drug in a quantification experiment, making the apparent absorbance of the drug larger than it actually is. This generates a systematic error which is particularly pronounced at low *NAR%*. This error existed in all cases where the measured spectroscopic signal of the drug overlapped with the absorption of the nanoparticle carrier. Then, the indirect approach was prone to errors in the entire concentration range of the drug, either due to incomplete separation of drug from the nanoparticles at low drug concentrations or due to dilution errors at high drug concentrations.

## 4. Conclusions

A straightforward, economical and fast method for drug quantification in nanoparticles was developed using ATR-FTIR spectroscopy of dry films. The method presents excellent linearity in the range of 0.00 to 7.89 *NAR%*, despite the low drug concentrations. The *DLC*_LOD_ and *DLC_LOQ_* were 0.3 and 1.0%, respectively, for NAR in SFN. Reproducibility in terms of relative standard deviation after measuring 10 blank samples was 2.0%. The method was shown to be robust, as calibration lines measured in two different institutions showed no statistical differences.

The methodology developed here is less susceptible to over- or under- estimation of *DLC* which can occur using conventional direct and indirect approaches [30,31], and can be easily extrapolated to other drug-loaded SFNs and even to other polymeric nanocarriers. Furthermore, it can operate on the final drug product without requiring access to the loading solution. Here, it is faster than the direct approach because it does not require drug extraction and phase separation.

In the present case, a simple evaluation of the absorbance at one particular wavenumber was sufficient for quantifying a drug in protein nanoparticles. Other drugs and carrier systems might present more challenging cases and require tailored approaches for evaluation. These could consist of considering (i) the signal at several wavenumbers to eliminate the protein contribution to the measured signal, (ii) the second derivative of absorbance for a better separation of drug and protein bands, and (iii) the use of multivariate statistical methods to identify relevant spectral regions and to establish calibration models. However, in all cases, it would be beneficial to base the quantification on drug bands which are little affected by the environment, i.e., bands which are caused by vibrations with small contributions from functional groups which are able to interact with the protein matrix. For this reason, we used the band of a skeletal aromatic vibration at 1085 cm^−1^ and disregarded the stronger OH band at 1156 cm^−1^, which shifted when the drug was incorporated into the protein matrix. The same applies to the bands of the protein matrix that are used for normalization. They should not be affected by interactions with the drug.

Using the second derivative has the advantage of better resolving overlapping bands, as mentioned above, but also presents challenges, as it requires high quality spectra. Here, the liquid nitrogen cooled Mercury-Cadmium-Telluride detector performed much better than the Deuterated Triglycine Sulfate detector, which can be operated at room temperature. Only a considerable increase of the spectrum recording time with the latter detector can mitigate the quality difference. In contrast, when absorbance spectra are analyzed, most modern IR detectors will work well in the mid-IR spectral range and should not be a concern to the user.

Another possible problem is water vapor, which might have a significant impact on the spectral regions 1250–2100 cm^−1^ and 3300–4000 cm^−1^ when using unpurged equipment. In the approach presented here, the water vapor regions were not used for the analysis and, thus, were not a concern. Nonetheless, we encourage users to be aware of this problem, as computational water vapor compensation might work for some applications but not for others. The water vapor problem will be considerably more severe when the second derivative of absorbance is analyzed, as this enhances the sharp water vapor bands relative to the broader analyte and protein bands.

## Figures and Tables

**Figure 1 pharmaceutics-12-00912-f001:**
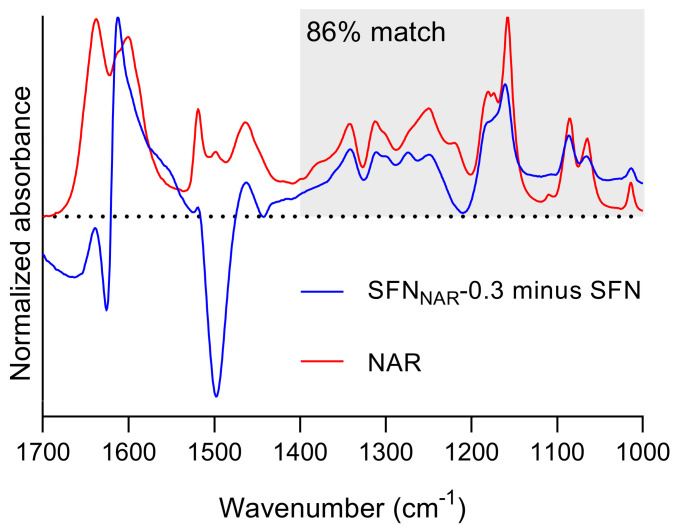
Specificity of ATR-FTIR spectroscopy for identifying drugs in silk fibroin nanoparticles (SFN). In blue, the resulting spectrum after subtraction of the SFN spectrum from the SFN_NAR_-0.3 spectrum. In red, the spectrum of NAR measured from a dried film. The gray rectangle shows the area used for matching.

**Figure 2 pharmaceutics-12-00912-f002:**
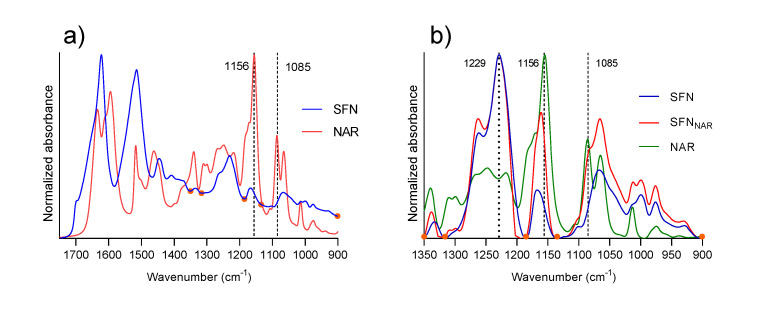
(**a**) Original infrared absorption spectra of SFN and NAR. (**b**) Base-line corrected infrared absorption spectra of SFN, NAR and SFN_NAR_-1.2. The SFN and the SFN_NAR_ spectra are normalized to an absorbance of 1 at the maximum of the amide III band at 1229 cm^−1.^ The NAR spectrum is normalized to an absorbance of 1 at 1156 cm^−1^. All samples were measured as a dry film obtained from a 30% EtOH solution. Orange dots indicate the baseline points.

**Figure 3 pharmaceutics-12-00912-f003:**
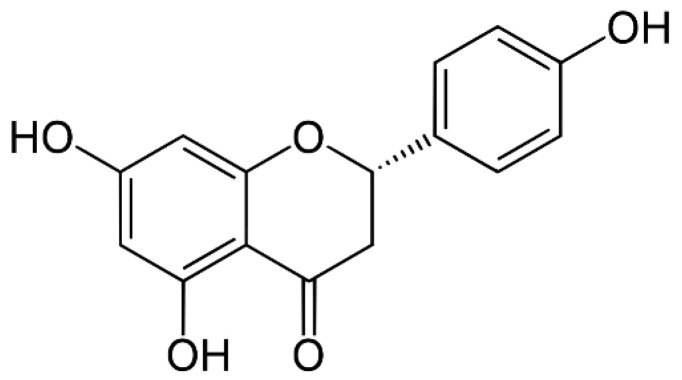
Structure of NAR.

**Figure 4 pharmaceutics-12-00912-f004:**
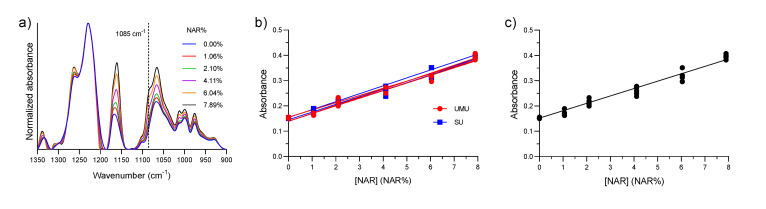
(**a**) Absorbance spectra of SFN with increasing quantities of NAR; (**b**) Six calibration data sets and their respective calibration lines, three obtained at the University of Murcia (UMU) and three at Stockholm University (SU). (**c**) Calibration data set including all data points.

**Figure 5 pharmaceutics-12-00912-f005:**
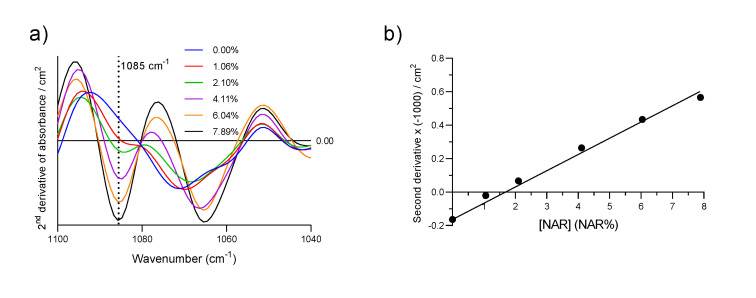
(**a**) The second derivative of the absorbance spectra of SFN with increasing quantities of NAR obtained at Stockholm University; (**b**) Calibration line constructed with the second derivative of the Absorbance spectra at 1085 cm^−1^ multiplied by −1. Each spectrum was smoothed with a thirteen-point Savitsky–Golay function before derivation.

**Figure 6 pharmaceutics-12-00912-f006:**
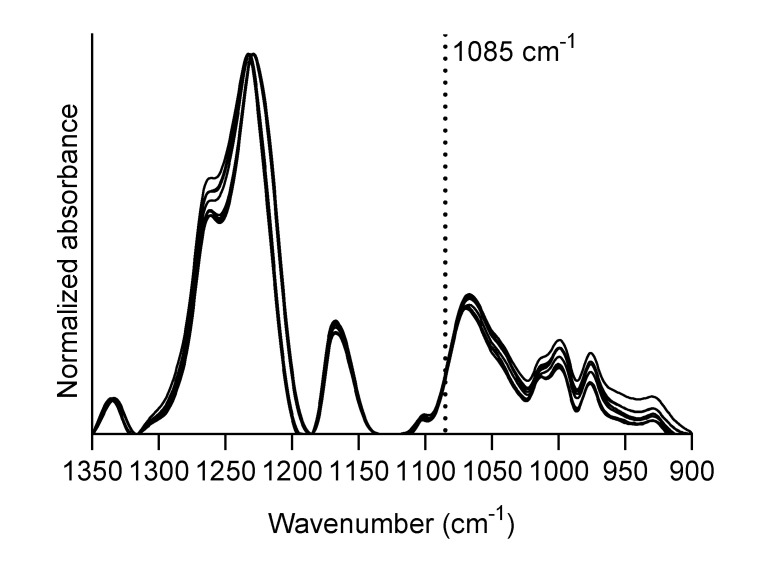
Baseline corrected and normalized spectra of the 10 blank SFN samples used for the calculation of limit of detection (*LOD*), limit of quantification (*LOQ*) and the reproducibility of the method.

**Table 1 pharmaceutics-12-00912-t001:** Calibration curve parameters. The first two columns list the average parameters and standard deviation (σ) for each set of three calibration lines obtained at the University of Murcia (UMU) and Stockholm University (SU). The third column lists the parameters of the calibration line fitted to all data points under the constraint of passing through the average value at 0 *NAR%*.

Fitting Parameters	UMU	SU	All Data Points
	Mean ± σ		
**Best-fit values**			
Slope *	0.030 ± 0.002	0.031 ± 0.001	0.029
Y-intercept	0.145 ± 0.009	0.146 ± 0.008	0.152
**Goodness of Fit**			
R^2^ **	0.978 ± 0.007	0.982 ± 0.019	0.973

* Slope units are *NAR%*^−1^, ** R^2^, coefficient of determination.

**Table 2 pharmaceutics-12-00912-t002:** Average (*Ā*)and standard deviation (σ) of absorbance at 1085 cm^−1^ used for the calculation of limits of detection (*A*_LOD_ and *DLC*_LOD_), limits of quantification (*A*_LOQ_ and *DLC*_LOQ_) and reproducibility of the method according to Equations (4)–(6).

Quantity	Value
Average of absorbance at 1085 cm^−1^ (*Ā*)	0.151
Standard deviation (σ)	0.003
Reproducibility (%)	2.0
*LOD* of absorbance (*A_LOD_*)	0.161
*LOQ* of absorbance (*A_LOQ_*)	0.181
*DLC_LOD_* (%)	0.3
*DLC_LOQ_* (%)	1.0

**Table 3 pharmaceutics-12-00912-t003:** Evaluation of the accuracy of the method for the determination of NAR in SFN.

Sample	Recovery (%)
1	103%
2	104%
3	102%
4	106%
Average	104%
Standard deviation	2%

**Table 4 pharmaceutics-12-00912-t004:** Drug loading content (*DLC*) and entrapment efficiency (*EE*) of NAR-loaded SFN are expressed as mean ± standard deviation. n = 3.

Sample	*DLC* (%)	*EE* (%)
	Mean ± σ (RSD)	
SFN_NAR_-0.3	0.87 ± 0.07 (8.7%)	41.0 ± 3.0 (7.3%)
SFN_NAR_-1.2	3.13 ± 0.08 (2.6%)	38.0 ± 1.0 (2.6%)
SFN_NAR_-4.5	4.06 ± 0.06 (1.6%)	13.2 ± 0.2 (1.5%)

RSD, relative standard deviation.

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
