# Peer review of "Direct Quantification of Drug Loading Content in Polymeric Nanoparticles by Infrared Spectroscopy"

_pharmaceutics, 2020, doi:10.3390/pharmaceutics12100912_

Round 1

Reviewer 1 Report

The paper introduces a method for the direct quantification of drugs in polymer nanoparticles using ATR FT-IR. This method is simple to operate and reliable in results. It will play an important role in the research of drug delivery systems in the future. The paper also selected two universities and two different equipment for calibration. The results showed significant similarities, which proved the robustness of the method. Here, I make two suggestions, hoping to help improve the recovery.

  1. Before doing baseline calibration, should the authors consider performing ATR calibration  first?
  2. Could the authors consider using the peak height of the secondary derivative spectra for quantification? Because the peak positioning is accurate in secondary derivative spectra. The secondary derivative spectra may also eliminates the effects of baseline and overlapping peaks to some extent.

Author Response

We thank the reviewer for taking the time to review our work and refer him/she to the attached document for the full response.

Reviewer 2 Report

The Authors developed a new, reliable and simple method for drug quantification in polymer nanoparticles. The method is based on attenuated total reflection Fourier transform infrared spectroscopy. This is the first time when this strategy has been applied for direct drug quantification in nanoparticles.

The manuscript is well written, and the data is clearly presented. The Authors explain in details their method and define the range of its applicability. Two independent verifications in different labs confirm the reliability and robustness of such a strategy for drug quantification.

I recommend this manuscript for publication in Pharmaceutics as is.

Author Response

We thank the reviewer for taking the time to review our work and refer him/her to the attached document for the full response.

Reviewer 3 Report

This manuscript describes an application of FTIR-ATR to quantification of drug loading, with a carefully executed method, easy to follow reasoning, carefully described analysis and fully supported by the data and as such I recommend publication after some minor editing.

I have some suggestions to improve the manuscript.

Firstly, while I fully accept the argument that the necessity to dilute samples for UVvis and the overlap between the silk and the small molecule spectra makes quantification from UV absorbance much more error-prone, the new generation of UVvis spectrophotometers using ATR can measure absorbances up to 300, so I think part of the argument is no longer as strong as it would have been a few years ago, and a slight rephrase of the introduction might be useful.

Using two different labs and two different IRs is part of the good method design in the manuscript but I'm not sure a simple user of FTIR will appreciate (or even be aware of) differences such as detector type and purging, especially since most instrument software uses automatic atmospheric compensation (thus hiding the problems from the user) and the ATR accessory itself in many instruments I have seen is not purged. It might be useful to point out that most IR detectors work well in the 400 to 4000 cm-1 region and that the user should choose atmospheric compensation.

To make the method more widely useful, it would be good to include Figure 1 in the discussion of interaction between drug molecule and carrier - it is not just the drug molecule spectrum which can experience spectral shifts, but the carrier as well and I would guess that some small changes in the structure of the silk lead to changes in amide I and amide II peak shapes. We have seen such shifts when proteins interact with fabrics, for example. Differences could perhaps also be caused by different batches of silk, Figure 5 shows differences in the amide III peak positions and it would be useful to at least point out that some carrier peaks can also shift and that anyone adopting the method should look out for both effects.  

Equations 2 and 3 have some erroneous words in there and show poor use of the equation editor. Depending on whether MDPI edits equations, I would recommend using italics for all or none of the text or perhaps using a variable such as m_NAR and defining that. Lines 403 to 405 should have been deleted as they seem to come from the MDPI template.

Author Response

(The authors gave the same response as above.)
